# A bioavailable $^{87}$Sr/$^{86}$Sr isoscape of Mongolia: Implications for the reconstruction of past human and animal mobility

**Mael Le Corre** [1,2]*, **Eric Douville**[3], **Arnaud Dapoigny**[3], **Khanh-Vi Tran**[3], **Tsagaan Turbat**[4], **Ganbold Enkhbayar**[5], **Sébastien Lepetz**[1], **Antoine Zazzo**[1]

**1** BioArchéologie, Interactions Sociétés Environnements (BioArch, UMR 7209), Muséum National d'Histoire Naturelle, Sorbonne Université, Centre National de la Recherche Scientifique (CNRS), Paris, France, **2** Laboratoire de Géologie de Lyon, Terre, Planètes, Environnement (LGLTPE, UMR 5276), École Normale Supérieure Lyon, Université Lyon 1, Centre National de la Recherche Scientifique (CNRS), Lyon Cedex, France, **3** Laboratoire des Sciences du Climat et de l'Environnement (LSCE/IPSL, UMR 8212), Laboratoire des Sciences du Climat et de l'Environnement, LSCE/IPSL, UMR CEA-CNRS-UVSQ, Université Paris-Saclay, Gif-sur-Yvette, France, **4** Institute of Nomadic Archaeology and Department of Anthropology and Archaeology, National University of Mongolia, Ulaanbaatar, Mongolia, **5** National Museum of Mongolia, Ulaanbaatar, Mongolia

\* mael.lecorre@abdn.ac.uk; mael.lecorre@mnhn.fr

## Abstract

Understanding past human and animal mobility is essential for reconstructing the social and cultural dynamics of ancient societies. Strontium isotope analysis ($^{87}$Sr/$^{86}$Sr) offers a powerful tool to investigate provenance and movement. The bioavailable $^{87}$Sr/$^{86}$Sr follows the underlying lithology, and increasing efforts have been made to map its spatial distribution across the landscape and produce $^{87}$Sr/$^{86}$Sr isoscapes at local and more global scales. Mongolia's steppe landscapes have long supported highly mobile pastoralist societies whose movements were central to the formation of major polities, including Late Bronze Age cultures, Xiongnu first nomadic state and Mongol Empire. Despite the growing use of $^{87}$Sr/$^{86}$Sr isotopic analysis to investigate past mobility, the lack of a robust $^{87}$Sr/$^{86}$Sr baseline has hindered large-scale interpretations. Here, we generated a regionally-calibrated bioavailable $^{87}$Sr/$^{86}$Sr isoscape for Mongolia by analysing modern plants collected at 534 sites, mostly from Arkhangai, central Mongolia. We used two machine learning approaches: the random forest regression (RF) and the ensemble machine learning (EML). Both methods produced comparable isoscapes with RF slightly outperforming EML. Three major regions have been identified, reflecting the complex geology of Mongolia: a western intermediate-to-high $^{87}$Sr/$^{86}$Sr region (0.710–0.722), a central low $^{87}$Sr/$^{86}$Sr region (0.707–0.711), and an eastern intermediate $^{87}$Sr/$^{86}$Sr region (0.709–0.712). Comparison of archaeological $^{87}$Sr/$^{86}$Sr data from 25 sites across Mongolia from the Late Bronze Age to the Medieval period with local isoscape predictions suggest that human, animal and plant samples are mainly of local origins for most of the sites,

**Data availability statement:** All relevant data are within the paper and its Supporting Information files. A link to download the covariates used in the RF and EML analyses is provided within the R script.

**Funding:** This research, including salary support for MLC and KVT, sample collection, and isotope analysis (LSCE), was funded by the French National Research Agency under grant number ANR-20-CE27-0018-02. The funders had no role in study design, data collection and analysis, decision to publish, or preparation of the manuscript. There was no additional external funding received for this study.

**Competing interests:** The authors declare that the research was conducted in the absence of any commercial or financial relationships that could be construed as a potential conflict of interest.

regardless of the period. This isoscape provides a strong baseline in Mongolia and not only enhances archaeological interpretations of past mobility but also holds significant potential for ecological monitoring and the authentication of regional products, such as cashmere.

## Introduction

Identifying the provenance of past humans and animals, and reconstructing their mobility is crucial for understanding the evolution of the past human populations. Mobility provides key insights about population movements, cultural and commercial exchanges, ritual and social practices, and societal dynamics that shaped past societies [1–4]. The radiogenic strontium isotope, and more particularly the isotopic ratio $^{87}Sr/^{86}Sr$, is a particularly powerful tool to study this mobility [5–7]. Mainly driven in the environment by the underlying lithology, the bioavailable $^{87}Sr/^{86}Sr$ (i.e., the $^{87}Sr/^{86}Sr$ available for absorption by living organisms) distribution is spatially well defined in the landscape [8,9]. Comparing $^{87}Sr/^{86}Sr$ from archaeological remains to regional/local $^{87}Sr/^{86}Sr$ allows discriminating between local and non-local individuals [7,3,10] and when the spatial distribution of $^{87}Sr/^{86}Sr$ is known in a given area, the origin of the samples can be assessed [11,12].

Strontium has one radiogenic isotope, the $^{87}Sr$, product of the radioactive decay of $^{87}Rb$ (rubidium). Abundance of $^{87}Sr$ in rocks depends on the initial $^{87}Rb$ content and the age of formation of any geological unit, affecting its $^{87}Sr/^{86}Sr$ value. Older $^{87}Rb$-rich rocks, such as granite or rhyolite, are expected to be highly radiogenic with elevated $^{87}Sr/^{86}Sr$ [5,9]. In contrast, younger rocks or those with low initial $^{87}Rb$ content, such as basalt or carbonate, are expected to have lower $^{87}Sr/^{86}Sr$ [5,9]. Strontium from bedrocks is released into soil and surface water through weathering and leaching, and constitutes the main contributor to the pool of $^{87}Sr/^{86}Sr$ available to plants and animals [5]. As a result, the spatial distribution of bioavailable $^{87}Sr/^{86}Sr$ across the landscape follows discrete patterns, mirroring the nature of the underlying bedrock, and remains stable at the human timescales [8]. Additional $^{87}Sr/^{86}Sr$ sources contribute less significantly to this pool, including rainfall, sea spray, atmospheric deposits or fertilizer in anthropogenic contexts [6,9]. Plants absorb the bioavailable $^{87}Sr/^{86}Sr$ directly from the soil and animals through their food and water intake [5]. Once ingested and absorbed in the gut, strontium is incorporated into bioapatite in place of calcium, with which it shares ionic properties, within hard tissues (e.g., bones, enamel, otoliths) during their formation or mineralization [13–16]. Fractionation of $^{87}Sr/^{86}Sr$ occurring during the process is negligible [15], allowing the direct inference of the local environmental $^{87}Sr/^{86}Sr$ from the $^{87}Sr/^{86}Sr$ measured in the tissues.

Knowing the distribution of bioavailable $^{87}Sr/^{86}Sr$ across the landscape makes it possible to determine the geographic origin of a sample and to reconstruct past mobility when time series of $^{87}Sr/^{86}Sr$ are available [11,12,14]. Three main approaches are employed to map the spatial distribution of bioavailable $^{87}Sr/^{86}Sr$ and to generate so-called bioavailable $^{87}Sr/^{86}Sr$ isoscapes from empirical data [17,18]. First, the

domain mapping approach involves dividing a given region into sub-areas defined by their differences in geological and lithological characteristics [19,20]. The bioavailable $^{87}Sr/^{86}Sr$ of each sub-area is directly estimated from soil, plant or animal data, collected within that specific region. The isotopic signatures of archaeological samples are then compared with the $^{87}Sr/^{86}Sr$ of the different sub-areas to determine whether they are local or to identify their origin [7,21]. The second methodology, the contour mapping isoscapes, relies on geostatistical methods to interpolate $^{87}Sr/^{86}Sr$ between sampling sites [17,22,23]. Methods such as kriging with external drift are used to account for the discrete spatial distribution of lithological units [22,24]. The $^{87}Sr/^{86}Sr$ is modeled as continuous distribution surfaces, associated with spatial uncertainty maps. These predictions and uncertainty maps enable probabilistic geographic assignments, estimating the likelihood of a sample's origin across the isoscape [17,25]. Relatively easy to apply, domain and contour mapping share however similar limitations. For both approaches the extent of isoscapes is constraint by the sampling coverage and they require extensive sampling effort to be well defined. [17,18]. The third category of methods, based on machine learning algorithms such as random forest regressions (RF) or ensemble machine learning (EML), has received increasing interest, aiming at producing isoscapes at very large scales, even in areas with limited or no sampling effort [17,18,26–29]. RF combines geological variables and auxiliary environmental sources of $^{87}Sr/^{86}Sr$ in the ecosystem to predict bioavailable $^{87}Sr/^{86}Sr$ alongside spatial uncertainty [17]. Bataille et al. [9] used RF to generate a global bioavailable $^{87}Sr/^{86}Sr$ isoscape, integrating their former mechanistic $^{87}Sr/^{86}Sr$ bedrock model, predicting $^{87}Sr/^{86}Sr$ of bedrock from bedrock age and $^{87}Rb/^{86}Sr$ content of actual and parent rock material [30], as well as other geological, climate and environmental variables. The model was trained on a worldwide bioavailable $^{87}Sr/^{86}Sr$ dataset, including water, soil, plant and animal samples, compiled by the authors [9,17]. RF demonstrates strong predictive power even in regions with low sampling coverage, provided that the geological and environmental conditions in these areas are well represented in the training dataset [9]. Otherwise, the model lacks in accuracy, requiring additional sampling for local calibration [9,26,31]. As an improvement over the RF approach, EML relies on multiple algorithms, including RF, rather than a single model. By aggregating the predictions of these different algorithms, trained individually on the same dataset, EML aims to reduce bias and variance, leading to more reliable and precise predictions [26,32]. Applied to generate a bioavailable $^{87}Sr/^{86}Sr$ isoscape of Eastern Canada and integrating geographic features to account for spatial autocorrelation between sampling sites beside geological and environmental variables, EML predicted similar spatial distribution of $^{87}Sr/^{86}Sr$ compared to RF but with improved spatial uncertainty, particularly in highly radiogenic regions [26].

The spatial distribution of bioavailable $^{87}Sr/^{86}Sr$ in Asia, notably in High Asia (Tibetan Plateau and surrounding mountain ranges), Central Asia (Kazakhstan and neighboring steppe countries) and Mongolia, is poorly documented despite the fact that questions related to human mobility are central to understanding transformation in past Asian societies. As a vast steppe-dominated territory, Mongolia has historically been home to highly mobile pastoralist societies whose movements played a fundamental role in the emergence and expansion of key societies, including the Late Bronze Age (LBA) cultures, the Xiongnu Empire (c. 3rd century BCE–1st century CE), and the Mongol Empire (13th–14th centuries CE). The emergence of nomadic pastoralism in Inner Asia is generally situated in the Late Bronze Age [33]. This development is commonly associated with the introduction of the domesticated horse in the region [34]. However, the precise conditions of this transition and especially the organization of human mobility remain largely undocumented (e.g., short- vs. long-distance movement, frequency, cyclicity). This major economic shift was accompanied by profound transformations in ritual and social practices, and in their material expression, as seen in complex funerary monuments [32]. Similar questions regarding agro-pastoral practices, mobility, and the development of cultural and commercial interaction networks emerged in later periods as well—particularly during the Xiongnu era, when the Xiongnu established the first nomadic state in Mongolia [33], and throughout the Genghisid period, during which mobility was a critical factor underpinning the unprecedented expansion and consolidation of the Mongol empire [35]. Both contexts illustrate increasing complexity in territorial organization, systems of control, and socio-political structures. Mobility of past Mongol populations has been investigated through $^{87}Sr/^{86}Sr$ isotopic analyses, mostly for the Late Bronze Age (e.g., [36–38]) and the Xiongnu period (e.g., [39–41]).

However, the absence of a well-established $^{87}Sr/^{86}Sr$ baseline (isoscape) for Mongolia has significantly limited interpretations and did not allow for broader inferences about large-scale movement patterns.

While the global isoscape from Bataille et al. [9] encompasses Mongolia and provides the first $^{87}Sr/^{86}Sr$ isoscape of the country, the model was trained on very little data from Asia and none from Mongolia, making predictions for this region less accurate. More recently, at a very local scale, a contour mapping isoscape was produced using new plant data, within the Altai region, in the extreme west of Mongolia, but its application remained restricted to local mobility studies [23]. Then, by integrating the plant data used to build this local isoscape with the global $^{87}Sr/^{86}Sr$ dataset [9], an updated and extended isoscape was generated using RF [38], theoretically covering all of Mongolia. However, despite reliable predictions at the scale of the Altai Mountains, the authors cautiously limited their interpretations of mobility and origin beyond the Altai region, as the calibration of the isoscape for the rest of the country was done on data from a very limited area. Consequently, no bioavailable $^{87}Sr/^{86}Sr$ isoscape calibrated currently exists for Mongolia. Finally, our study aims to fill this gap. Through an extensive sampling campaign in central Mongolia, supplementing the global dataset [9,26], we developed a regionally calibrated bioavailable $^{87}Sr/^{86}Sr$ isoscape for the Arkhangai region, extending it to the entire Mongolian territory. This sampling campaign was part of a broader research project aiming to understand the role of horses in the interactions among the Late Bronze Age pastoral communities of central Mongolia, a region with an exceptional concentration of ancient funerary structures [42–44]. Based on machine learning algorithms, the isoscape was generated using both the traditional RF approach and the EML approach to assess how EML improves predictions. Finally, we compared predicted local $^{87}Sr/^{86}Sr$ from the isoscape with published $^{87}Sr/^{86}Sr$ from archaeological samples, primarily human, animal and plant from the LBA and the Xiongnu periods, alongside a smaller set of modern samples, all collected from multiple sites across Mongolia, to discuss their mobility patterns across the country in a diachronic perspective.

## Materials and methods

### Study area

Mongolia has an average elevation of 1500 m, with some regions exceeding 4000 m. Mongolia's landscape transitions from arid desert and semi-desert in the south, spanning from the Govi-Altai aimag (first-level administrative division of Mongolia) in the west to the Dornogovi aimag in the east (Fig 1), to expansive steppes and forest steppes in the northern half of the country. Three major mountain ranges shape Mongolia's topography: the Mongolian Altai mountains in the west and southwest of the country, the Khangai mountains in central Mongolia, and the Sayan mountains in the north, near the Russian border. The Arkhangai region encompasses part of the Khangai Mountains in the south with elevation gradually decreasing along a southwest–northeast axis. This area is predominantly covered by forest steppe, reflecting a transitional ecological zone between the mountainous terrain and the open steppes. The Arkhangai region is archaeologically very rich with a high concentration and a wide variety of monuments, mainly from Late Bronze Age [42–45]. Besides petroglyphs, graves and deer stones, khirgisuurs are a major component of the Arkhangai mortuary landscape. These structures consist of a central stone mound covering a burial chamber and surrounded by a fence of stones. They are associated with external stone circles and stone mounds, the latest covering horse's heads. Largest khirgisuurs include hundreds of stone mounds, providing an unrivaled source of material to understand the rise of nomad pastoralism during Late Bronze Age, through the reconstruction of horse mobility [42,45,46].

Mongolia exhibits a complex geology resulting from successive orogenic events (S1 Fig in S3 File). As part of the Central Asian Orogenic Belt, Mongolia is located between the Precambrian Siberian Craton to the north and the Tarim and Sino-Korean Cratons to the south [47]. The country can be divided into two primary geological domains: the Neoproterozoic – Early Paleozoic Northern domain and the younger Late Paleozoic Southern domain [47]. The northern part of Mongolia is dominated by ancient Archean to Proterozoic granitic and metamorphic rocks [47,48]. Mountain ranges formed during Paleozoic by the collision of micro-continents and island arcs are primarily composed of marine sedimentary rocks and magmatic rocks reflecting the subduction-related process and oceanic closure between continental blocks [47,48].

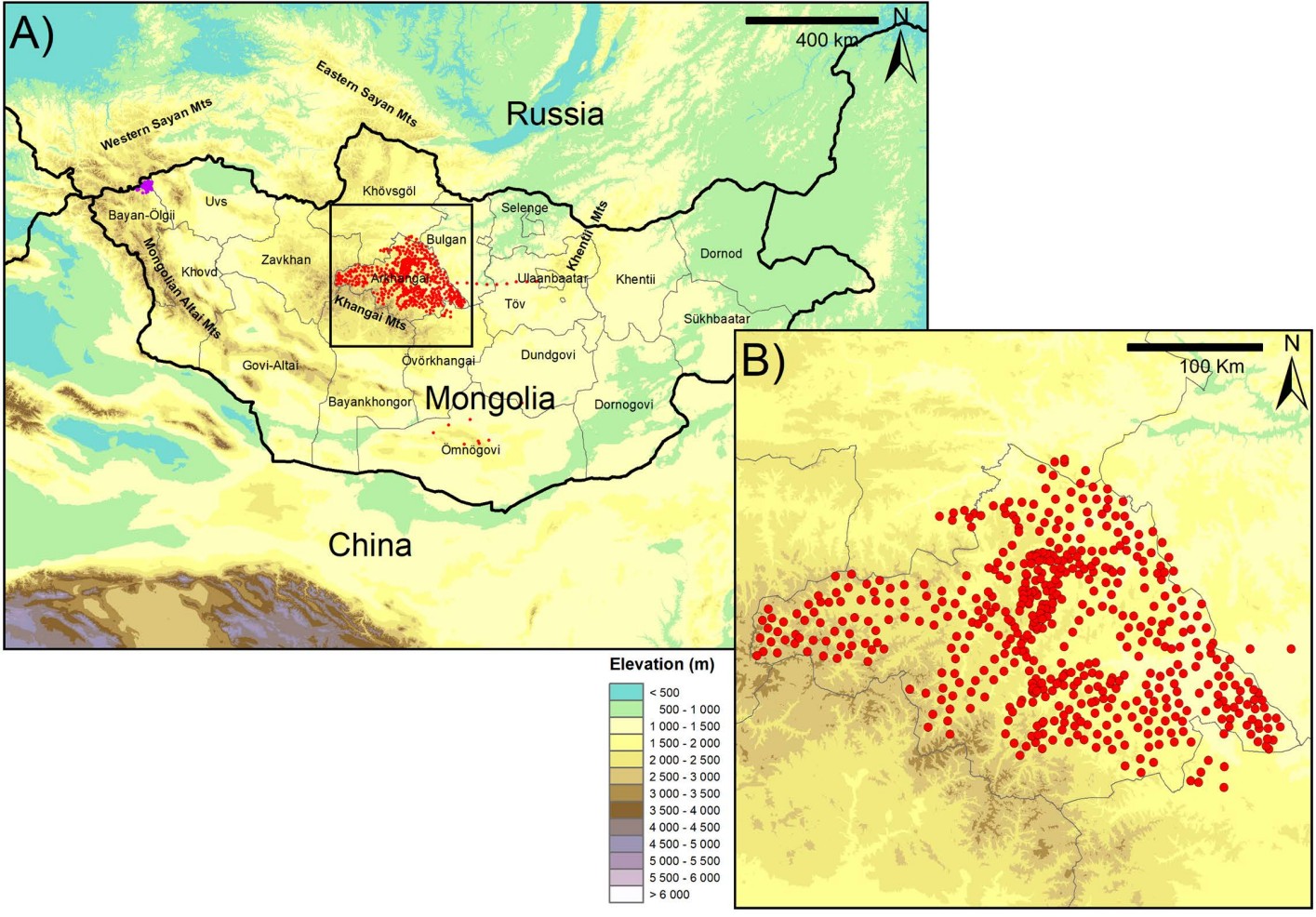

**Fig 1. Elevation map of Mongolia (A) with a focus on the Arkhangai regi on (B).** Locations of the plant sampled for this study are indicated in red and in a previous study in the Altai region [23] in purple. Basemap layer: digital elevation map SRTM 30m (NASA Shuttle Radar Topography Mission [49] downloaded from NASA Earthdata: https://search.earthdata.nasa.gov/).

More recent Cenozoic terrigenous sediments overlay these older formations. Between the Altai and the Khangai mountain ranges, a large Jurassic – Cretaceous sedimentary basin extends eastward into the Gobi Desert [47,48]. Another carbonate-rich sedimentary basin from Devonian –Permian is found East of the Khangai mountains. Finally, in the eastern part of the country, the Khentii mountains are dominated by a mix of Paleozoic and Mesozoic metamorphic and plutonic rocks [47,48].

## Plant sampling and mass spectrometry analyses

To generate the bioavailable $^{87}Sr/^{86}Sr$ isoscape for Mongolia, we completed the global bioavailable $^{87}Sr/^{86}Sr$ dataset from Bataille et al. [9], updated by Le Corre et al. [26], with recently published bioavailable $^{87}Sr/^{86}Sr$ data from Asia [23,50,51] and local $^{87}Sr/^{86}Sr$ plant values from the Arkhangai province (S1 Table). Plant samples (n = 513) were collected during three two-weeks campaigns (in 2019, 2020 and 2021) at 511 locations across the Arkhangai province over a 65 000 km² area (Fig 1B). Additionally, plant samples (n = 23) were opportunistically collected at 23 sites outside Arkhangai in 2023

and 2024. At each site, the sampled plants, collected within 2 meters of the GPS coordinates, consisted of a mix of available species, mainly dominated by grasses. Plant material (leaves, stem, seeds) were then dried and stored within sealed envelopes.

Plants samples were prepared at the Muséum national d'Histoire naturelle (MNHN, Paris, France) and at the Laboratoire des Sciences du Climat et de l'Environnement (LSCE, Saclay, France). For each sample, ca. 500 mg of dried plant material was crushed using a grinding mill then reduced in ashes using aluminum crucibles in a muffle furnace by applying a temperature cycle reaching 550°C during 12h. Chemical pretreatment and mass spectrometry were conducted at the LSCE in dedicated clean rooms. Plant ash was dissolved using 15 N $HNO_3$ for 2h then 30% $H_2O_2$ for 2h in a Teflon beaker and the solution was evaporated on a hotplate at 75°C. Dried samples were then dissolved using 4 mL of 3.5 M $HNO_3$ and loaded onto a column filled with the resin Sr-spec (Eichrome Inc.). Strontium was eluted from the resin using 2.5 mL of ultrapure Milli-Q water. Quality of the chemistry, including Sr extraction from the matrix and purification of interfering elements like Rb or REE, and Sr contents were systemically verified using a LSCE's ICPMS. Concentration of purified strontium solutions were adjusted to 20 ppb by dilution with 0.5 N $HNO_3$. The precise measurement of $^{87}Sr/^{86}Sr$ ratio was then carried out at the LSCE using the PANOPLY's Multi-Collector Inductively Coupled Plasma Mass Spectrometer Neptune$^{Plus}$ (MC-ICP-MS, Thermo Fisher Scientific) and following the analytical method recently updated at LSCE [52]. Corrections for all $^{87}Sr/^{86}Sr$ measurements accounted for Rb and Krypton (Kr) interferences. Standard-sample bracketing method was used to correct $^{87}Sr/^{86}Sr$ which was then normalized to the NBS 987 standard value of 0.710245. $^{87}Sr/^{86}Sr$ is reported with a 2σ uncertainty (S1 Table). Replicate analysis of the standard NBS 987 (National Institute of Standards and Technologies) was used to determine the typical reproducibility of strontium isotopic measurements with an overall mean value for this study of 0.710245 ± 0.000013 (± 18 ppm at 2σ, n = 238). Using home-made standard of plants (a few grams of hay) prepared many times during the period of measurement, the external reproducibility was better than 300 ppm (2σ, n = 10).

## Auxiliary variables

In order to predict the spatial distribution of the bioavailable $^{87}Sr/^{86}Sr$ across Mongolia, we used covariates that are likely to influence the bioavailable $^{87}Sr/^{86}Sr$ in the environment [9]. These covariates consist of lithological variables, topography, climate variables, soil characteristics, and atmospheric deposition [9,53]. Lithological variables notably included the predicted $^{87}Sr/^{86}Sr$ (median, 1$^{st}$ quartile, 3$^{rd}$ quartile) of the bedrock modeled from the age and nature of the lithological units [9,30]. Bedrock age and nature are the main driver of the $^{87}Sr/^{86}Sr$ variations in the landscape [8,9]. This predictive $^{87}Sr/^{86}Sr$ bedrock model integrates the main sources of geochemical variation that is expected to propagate into soils and consequently into the bioavailable $^{87}Sr/^{86}Sr$ pool [9,30]. All variables are detailed in Table 1 [9,11,48,49,54–61]. A link is provided in the R script to download the variables in raster format.

From the updated global $^{87}Sr/^{86}Sr$ database, we selected sampling sites with plant, soil or rodent data in order to ensure local $^{87}Sr/^{86}Sr$ values. Covariates data were then extracted at each sampling site location. When no data for a given covariate was available at a site due to the accuracy, resolution or coverage of the source data, we used instead the nearest value of this covariate in the vicinity of the site.

## Machine learning approach

Two machine learning approaches were used to generate the bioavailable $^{87}Sr/^{86}Sr$ isoscape: random forest regression (RF, [62]) and ensemble machine learning (EML, [63]). RF relies on regression trees, growing a "forest" of trees by bagging: for each tree, a training set is generated from the dataset by bootstrap sampling, and the left-over observations, or "out-of-bag", are used as a validation set. The tree is grown on the training set using a limited number of predictors randomly selected among the full set of predictors and the "out-of-bag" set is used for internal cross-validation. The outcome of each tree is then aggregated to obtain the predictions. No assumptions on data distribution and variance homogeneity

**Table 1. Auxiliary variables used in the random forest analysis.**

| Variables | Description | References |
|---|---|---|
| r.m1 | median bedrock model | [9] |
| r.srsrq1 | 1st quartile bedrock model | [9] |
| r.srsrq3 | 3rd quartile bedrock model | [9] |
| r.meanage_geol | mean GLiM age (Myrs) | [48] |
| r.minage_geol | minimal GLiM age (Myrs) | [48] |
| r.maxage_geol | maximal GLiM age (Myrs) | [48] |
| r.age | terrane age (Myrs) | [56] |
| r.mat | mean annual temperature (°C) | [55] |
| r.map | mean annual precipitation (mm.yrs$^{-1}$) | [55] |
| r.pet | global potential evapo-transpiration | [54] |
| r.ai | global aridity index | [54] |
| r.salt | Simulation of sea salt deposition (g.m$^{-2}$.yr$^{-1}$) | [57] |
| r.dust | dust deposition (g.m$^{-2}$.yr$^{-1}$) | [57] |
| r.fire | Black carbon deposition (kg.m$^{-2}$.s$^{-1}$) | [57] |
| r.foss | Fossil Fuel (kg.m$^{-2}$.s$^{-1}$) | [57] |
| r.volc | Volcanic deposition (kg.m$^{-2}$.s$^{-1}$) | [58] |
| r.dist | Distance to the coast (km) | [11] |
| r.elevation | shuttle radar topography mission (m) | [49] |
| r.bouguer | bouguer anomaly | [59] |
| r.GUM | global unconsolidated sediment map | [60] |
| r.cec | cation exchange capacity (mmol(c)/kg) | [61] |
| r.ph | soil pH (H2O, x10) | [61] |
| r.phkcl | soil pH (KCl, x10) | [61] |
| r.clay | clay (g/kg) | [61] |
| r.ocs | organic carbon stocks (t/ha) | [61] |
| r.bulk | bulk density (cg.cm-3) | [61] |

The bedrock model [9] estimates the distribution of predicted $^{87}$Sr/$^{86}$Sr values within lithological units based on their age and type [30]. GLiM: Global Lithological Map.

are required in RF [62]. RF shows very low sensitivity to collinearity between covariates [64] but a high correlation between two predictors can artificially amplify their influence on the model [65]. As such, we removed strongly correlated variables (R > 0.9) from the predictor set. Then, we conducted a variable selection step using the *VSURF* R package [66]. The selection relies on a three-step algorithm detecting and removing redundant and irrelevant predictors in RF models. The final model was run on the retained predictors, setting the number of trees be grown to 3000. The performance of the model was evaluated using the root-mean-square error (RMSE) and a ten-fold cross validation repeated five times. Variable importance was assessed using node impurity estimates, a measure of the efficiency of the trees to split the training dataset into two groups at each node. The relationship between the different covariates and the predicted $^{87}$Sr/$^{86}$Sr was visualized with partial dependence plots. The spatial uncertainty map associated with the $^{87}$Sr/$^{86}$Sr prediction map was generated using quantile random forest regression, with the standard deviation estimated from the 68.27% prediction interval [67]. Finally, in order to evaluate the efficiency of our sampling effort on local predictions, we ran 11 RF adding 0% to 100%, with a 10% increment, of the samples collected in the Arkhangai province to the global dataset [26]. Samples were selected randomly, and RMSE and spatial error were calculated on the remaining sampling sites, except when 100% of the samples were used. The process was repeated ten times.

EML refers to a collection of methods that integrate the predictions of multiple individual models (i.e., learners), to minimize the error and enhance the prediction accuracy [68]. The EML stacking approach, used in this study, aggregates predictions through the use of a meta-learner: a set of learners, based on different algorithms, are trained independently on the same dataset and their predictions are then used to train the meta-learner in order to generate the EML estimates [68]. We applied this approach to our dataset using the *landmap* R package [63]. The *landmap* function uses oblique geographic coordinates as covariate to account for spatial auto-correlation between samples [69]. Local auto-correlation may be weak due to sharp boundaries between geological units, with predictions relying mainly on environmental predictors. At broader spatial scales (regional or continental), however, geological units extend over large areas, and spatial dependency remains informative for the model [26]. Using oblique coordinates as covariate help capture these potential broad-scale spatial trends. Moreover, the modeling framework accounts for spatial dependencies by applying a spatial cross-validation approach to mitigate geographic sampling biases. We applied the EML on the same dataset as for the RF, using the predictors retained after the variable selection step. The model was trained using the default learners set of *landmap*: five base learners (RF, gradient boosting, support vector machines, neural networks, Lasso and elastic-net regularized generalized linear models), and a linear model for the meta-learner. A five-fold cross validation is used to assess the EML performance and *landmap* integrates the quantile random forest regression algorithm to compute the spatial uncertainty. From the model we generated the mean prediction map and the associated error map.

The RF analysis was conducted in R 4.4.0 [70] and the EML analysis was done in R 4.3.0 [71], as the *landmap* package was not available for more recent R versions. The R scripts, adapted from [26], are available as supplementary material SM4 in S4 File. The RF and the EML isoscapes with their associated uncertainty maps are available respectively supplementary material SM5 and SM6 in S5 File and S6 File.

### Archaeological $^{87}Sr/^{86}Sr$ data from the literature

We compiled the strontium isotope data from 17 studies conducted on 25 archaeological sites in Mongolia (S2 Table, [36–41,45,72–80]), with human (20 sites), animal (9 sites) and plant samples (1 site), mainly from Late Bronze Age and Xiongnu period. For a given site, data coming from different studies were pooled and we calculated a mean $^{87}Sr/^{86}Sr$ for human, animal and plant samples. We either used the coordinates of the archaeological site when provided or we approximated its location from the map provided in the publication. In the case of a site with multiple excavations (e.g., Baga Gazaryn Chuluu, Khösvgöl), we chose an average location. For each site we extracted the range of $^{87}Sr/^{86}Sr$ predicted by the isoscape within a radius of 10, 20 and 50 km. Several studies also provided modern animal and plant data (5 and 2 sites respectively, S2 Table, [36,37,40,76,79]) collected at or in the vicinity of the archaeological site. These data were used to validate the predictions of the isoscape at the sites.

## Results

### Model performance

The bioavailable $^{87}Sr/^{86}Sr$ isoscape for Mongolia is presented in Fig 2. Relying on published $^{87}Sr/^{86}Sr$ data and the plant samples collected for this study, the RF has a good performance with an RMSE of 0.0033 and a $R^2$ of 0.70. Variables retained in the model (Fig 3) were geological variables (1st quartile of the $^{87}Sr/^{86}Sr$ bedrock model, terrane age, minimum and maximum age of the lithological units), climate variables (mean annual temperatures and precipitation), atmospheric deposition (dust, volcanic, fire, fossil) and distance to the coast. The $^{87}Sr/^{86}Sr$ from the plants sampled for this study showed a very good agreement with the predictions of the isoscape at the sampling sites ($R^2=0.86$, Fig 4). The EML has similar performance compared to the RF (RMSE$=0.0034$, $R^2=0.71$). However, while relatively high, the correlation between predicted and observed $^{87}Sr/^{86}Sr$ values at the sampling sites was lower than for the RF ($R^2=0.78$).

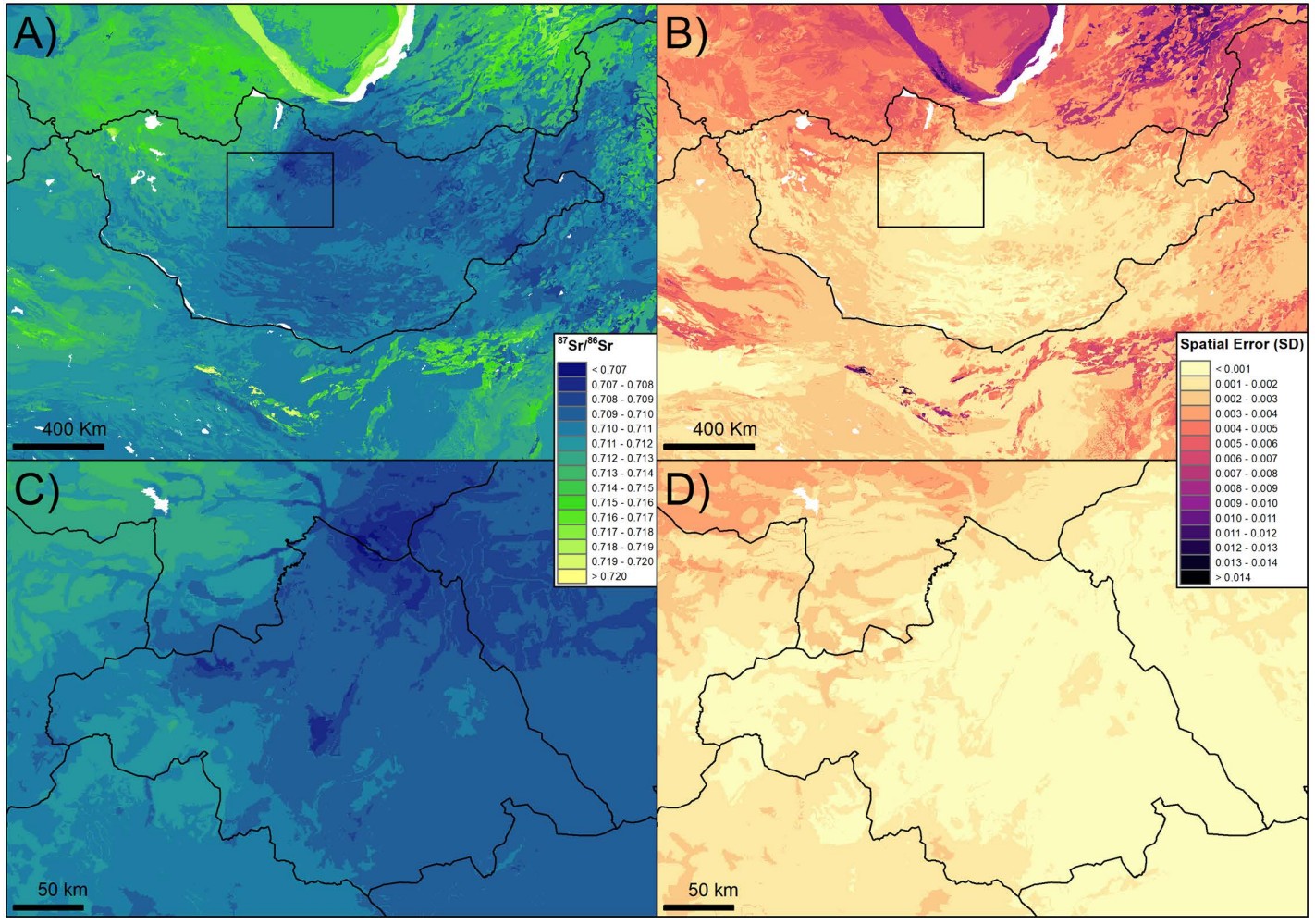

**Fig 2. Bioavailable $^{87}$Sr/$^{86}$Sr isoscape of Mongolia (A) and focus on Arkhangai (C) with the associated spatial uncertainties (B, D).**

When compared to a RF model built only on the already published data, without the samples collected for this study, adding 10% of the new sampled plants to the database drastically improved the prediction of the isoscape by reducing the RMSE from 0.0031 to 0.0010 when comparing measured $^{87}$Sr/$^{86}$Sr in plant to predict $^{87}$Sr/$^{86}$Sr at sampling sites (Fig 5A), and reduced spatial error at sampling sites from 0.0033 SD to 0.0007 SD (Fig 5B). Additional 10% increases of the number of new samples integrated to the bioavailable dataset marginally improved the RMSE and spatial error but reach RMSE = 0.0003 and spatial error = 0.0004 SD once all the new samples are added.

## Isoscape description

Modeled bioavailable $^{87}$Sr/$^{86}$Sr ranges in Mongolia from 0.707 to 0.722. Three main regions can be identified: the western region dominated by high $^{87}$Sr/$^{86}$Sr region (> 0.712), the central region dominated by low $^{87}$Sr/$^{86}$Sr region (<0.710), and the eastern region dominated by intermediate values (0.710–0.711). More precisely, the highest values are observed in the northwest aimags (Bayan-Olgii, UVS, Zavkhan, Khövsgöl) but also the highest spatial error, exceeding 0.005 in the Khövsgöl aimag. Southwest Mongolia (Govi-Altai and Bayan-Khongor aimags) presents intermediate $^{87}$Sr/$^{86}$Sr

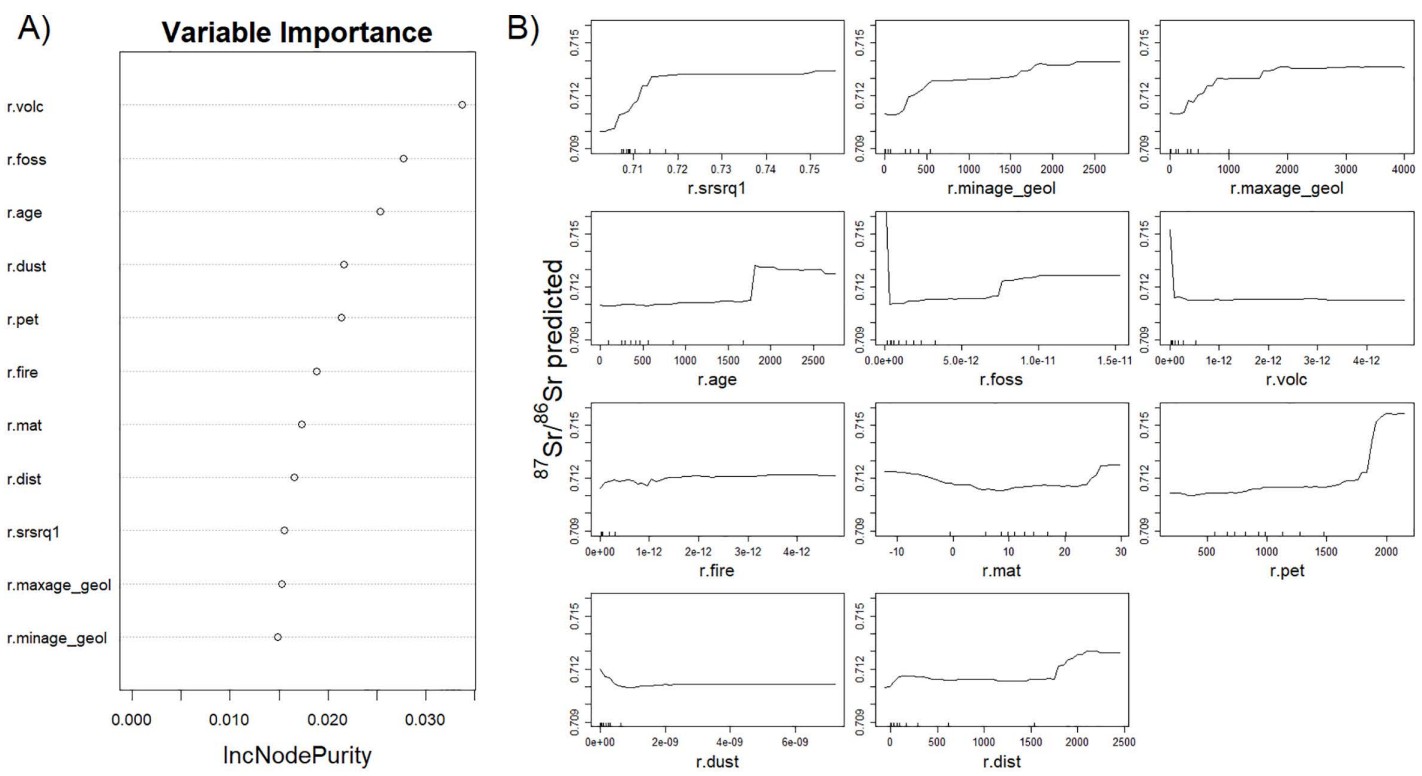

**Fig 3. Variable importance (A) and partial dependence plots (B) of the random forest regression.** Partial dependence plots depict the relationship between the predictors retained for the final model and the predicted bioavailable $^{86}$Sr/$^{87}$Sr.

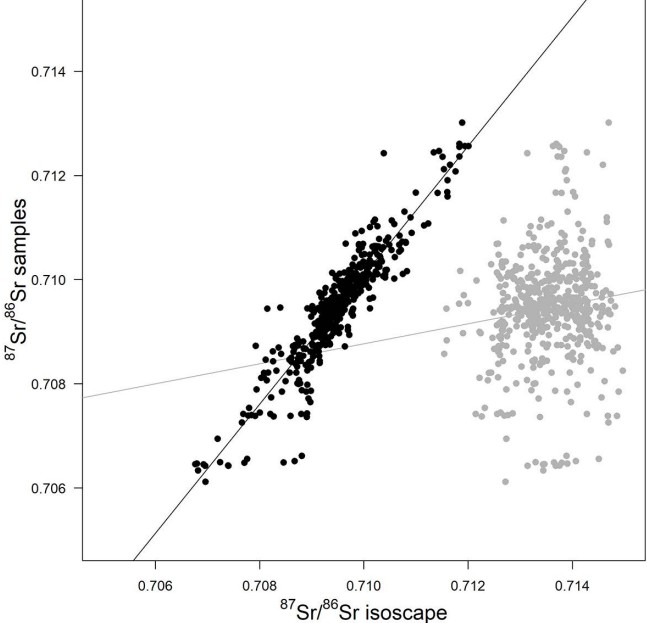

**Fig 4. Correlation between the $^{87}$Sr/$^{86}$Sr of the 536 plants collected for this study and predicted bioavailable $^{87}$Sr/$^{86}$Sr from the isoscape at the sampling sites.** Predictions from the locally-calibrated isoscape are shown in black. The predictions of the isoscape from Zazzo et al. [38] that do not include the newly collected samples are displayed in gray.

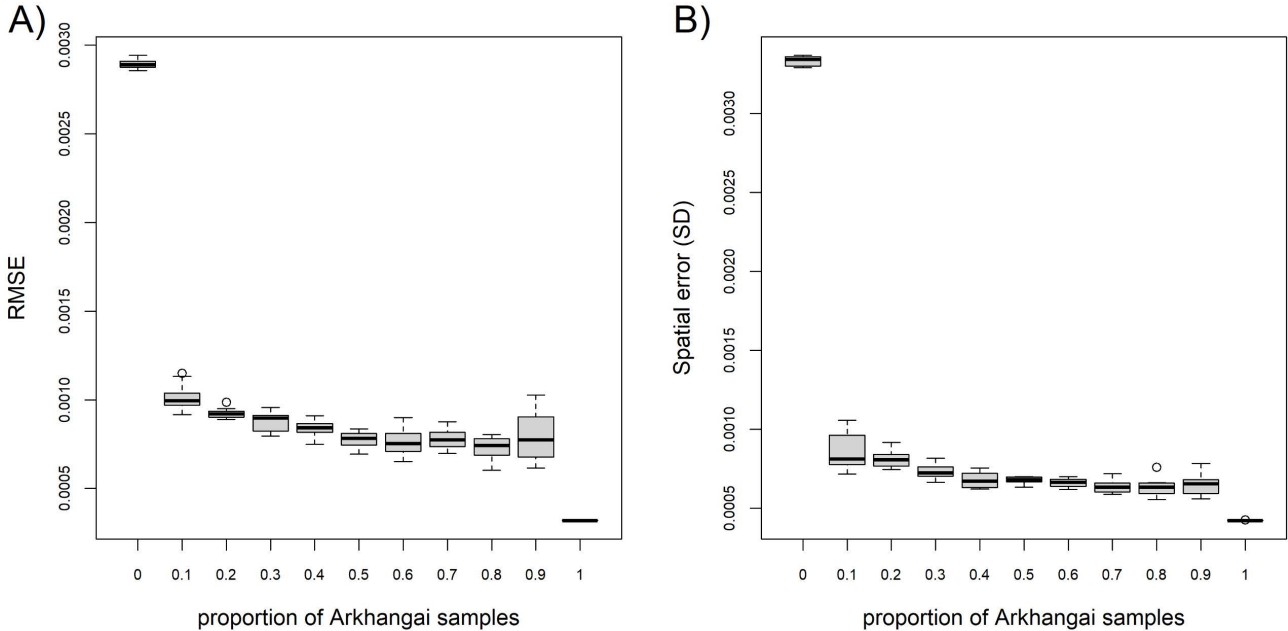

**Fig 5. Accuracy (A) and precision (B) of the RF local predictions at the sampling sites according to the proportion of newly collected plants added to the global dataset.** The root-mean square errors (RMSE) and standard deviation (SD) were calculated using sampling sites that were set aside, except for 100% where the RMSE was calculated using all the sampling sites.

(0.710–0.713), while lowest values are observed in central Mongolia, notably in the Bulgan, Tuv and Selenge aimags in the north (0.707–0.710). This decrease in $^{87}$Sr/$^{86}$Sr is coupled with a decrease in spatial error falling below 0.001. In the east, a mix of low and intermediate $^{87}$Sr/$^{86}$Sr is observed (0.709–0.712) with spatial error gradually increasing eastward, up to almost 0.005 SD in the Dornod aimag. In Arkhangai the $^{87}$Sr/$^{86}$Sr ranges from 0.707 to 0.712. A decrease in $^{87}$Sr/$^{86}$Sr can be observed from the southwest, in the Khangai Mountains, to the northeast at the junction of the Arkhangai, Khövsgöl and Bulgan aimags, mirroring the decrease in elevation (Fig 2). Spatial error is mainly below 0.001 but almost reaches 0.003 in some areas of the Khangai mountains. North of Mongolia, in Russia, Sayan mountains, Yablonoi mountains and Stanovoy Range are highly radiogenic regions with $^{87}$Sr/$^{86}$Sr ranging from 0.714 to 0.722 associated with relatively high spatial error. In China, the Xinjiang region, southwest of Mongolia, presents $^{87}$Sr/$^{86}$Sr ranging from 0.710 to 0.718. Bordering Mongolia in the south and east, the Inner Mongolia presents intermediate (0.710–0.712) to low values (0.708–0.710) with $^{87}$Sr/$^{86}$Sr locally rising above 0.712 in areas dominated by magmatic rocks (S1 Fig in S3 File).

The isoscape obtained with the EML displays similar, while more homogeneous broad patterns (Fig S2a, c in S3 File), with most of the prediction falling within +/- 0.001 of the RF isoscape $^{87}$Sr/$^{86}$Sr values (Fig S2e in S3 File). Main differences occur in the northwest, notably in the Khövsgöl aimag, with lower $^{87}$Sr/$^{86}$Sr and spatial error values and in the east, where both $^{87}$Sr/$^{86}$Sr and spatial error increase (Fig S2 in S3 File). These trends extend beyond Mongolia's borders. Spatial error and $^{87}$Sr/$^{86}$Sr are reduced in radiogenic regions of Russia covered by the isoscape. In eastern China, $^{87}$Sr/$^{86}$Sr is higher compared to the RF isoscape, but with a stable or lower spatial error (Fig S2 in S3 File).

## Comparison with archaeological data from archaeological sites

Most of the averaged $^{87}$Sr/$^{86}$Sr measured from the archaeological remains (21 sites out of 25) fall within the 50 km range of predicted $^{87}$Sr/$^{86}$Sr for the isoscape generated with RF (Fig 6). For 11 out of 25 sampled sites, mean values can be assigned within 10 km of the archaeological site. More precisely, among the 20 sites with human remains, 10 had a

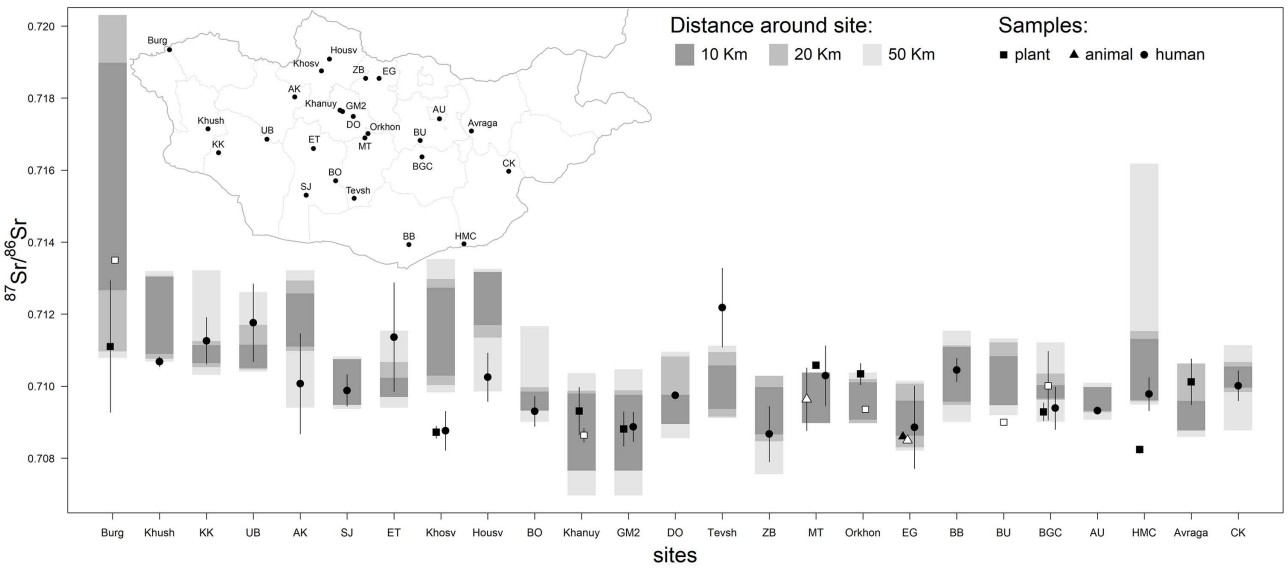

**Fig 6. Comparison between ⁸⁷Sr/⁸⁶Sr values of modern samples (plant, animals) and archaeological remains (human, animals) collected at 25 sites across Mongolia and the range of bioavailable ⁸⁷Sr/⁸⁶Sr predicted by the RF isoscape within 10 km, 20 km and 50 km of the archaeological sites.** Filled and open symbols correspond to archaeological and modern data respectively. The map shows the locations of the sites across Mongolia. Avraga: Avraga, AU: Agui Uul, AK: Avdalai Khyasaa, BGC: Baga Gazaryn Chuluu, Burg: Burgast, BB: Bayanbulag, BO: Bor Ovoo, BU: Bayan Unjuul, CK: Chandman Khar, DO: Dunde Orontso cemetery, Egiin Gol, ET: Emeelt Tolgoi, GM2: Gold Mod 2, HMC: Hets Mountain cave, Houvs: Houvsgol, Khanuy: Khanuy valley, Khosv: Khösvgöl (A, B, C, D, **F**), Khush: Khushuut, Khyar: Khyar Kharaach, MT: Maikhan Tolgoi, Orkhon: Orkhon valley (Moiltyn Am, Orkhon 1,7), SJ: Shine Jinst, Tevsh: Tevsh, UB: Ulaan Boom, ZB: Zuun Bel.

majority (> 50%) of human samples fitting with the local ⁸⁷Sr/⁸⁶Sr within 10 km of the sites and 16 within 50 km (Table 2). For 9 sites, all human samples fitted with the local ⁸⁷Sr/⁸⁶Sr range within 50 km (4 sites within 10 km). The Burgast site exhibits the highest heterogeneity in the isoscape prediction with local ⁸⁷Sr/⁸⁶Sr ranging from 0.709 to 0.720 within 50 km, followed by the Hets Mountain cave site (0.710–0.716). In contrast, the surroundings (< 50Km) of Shine Jinst, Tevsh, Maikhan Tolgoi, Orkhon valley, Egiin Gol and Agui Uul display the lowest variability with ⁸⁷Sr/⁸⁶Sr spanning less than 0.002. ⁸⁷Sr/⁸⁶Sr of samples from Kushuut and Maikhan Tolgoi (for animals, rodent samples) show a small deviation from the local values (< 0.0005). Only ⁸⁷Sr/⁸⁶Sr from samples at Khösvgöl, Tevsh and Hets Mountain cave (for animals, rodent samples), deviate significantly from the local baseline, showing differences greater 0.001.

Using the EML isoscape, the ⁸⁷Sr/⁸⁶Sr predicted within 50 km of the sites appeared less heterogeneous than those derived from the RF isoscape. Specifically, for 19 sites, the predicted values spanned a range of less than 0.002 (Fig S3 in S3 File). Consequently, the averaged ⁸⁷Sr/⁸⁶Sr of archaeological samples from 8 sites fall close but outside (deviation of < 0.0005) of the ⁸⁷Sr/⁸⁶Sr range predicted within 50 km of the site, whereas they fall within the 50 km ⁸⁷Sr/⁸⁶Sr range using the RF isoscape. Samples and local ⁸⁷Sr/⁸⁶Sr signatures remain different at Khösvgöl, Tevsh and Hets Mountain cave, but with values at Khösvgöl closer to the local baseline. However, at Kushuut, the samples that did not match the local ⁸⁷Sr/⁸⁶Sr range predicted by the RF isoscape, are consistent with the local values from the EML isoscape.

## Comparison with modern data from archaeological sites

Modern animal and plant samples were collected at 7 of the 25 archaeological sites. The averaged ⁸⁷Sr/⁸⁶Sr of these samples are consistent with the predicted local baseline within 10 km of Khanuy valley, Maikhan Tolgoi, Orkhon valley, Burgast and Baga Gazaryn Chuluu sites, and within 20 km of the Egiin Gol site (Fig 6). However, individual samples instead from Baga Gazaryn Chuluu, of the average ⁸⁷Sr/⁸⁶Sr, are only consistent with the local ⁸⁷Sr/⁸⁶Sr range within 50 km of the site

**Table 2. Proportion of archaeological and modern samples from Mongolian archaeological sites falling within the bioavailable $^{87}Sr/^{86}Sr$ range predicted by the isoscape within 10 km, 20 km and 50 km from the sites.**

| Site | Type | Period | n | Proportion of samples (%) within the $^{87}Sr/^{86}Sr$ range at | | |
|---|---|---|---|---|---|---|
| | | | | 10km | 20km | 50km |
| **Archaeological samples** | | | | | | |
| Agui Uul | human | EIA | 1 | 100.00 | 100.00 | 100.00 |
| Avdalai Khyasaa | human | LBA | 4 | 25.00 | 25.00 | 75.00 |
| Avraga | animal | ME | 10 | 30.00 | 70.00 | 70.00 |
| Baga Gazaryn Chuluu | animal | Xiongnu | 4 | 25.00 | 25.00 | 100.00 |
| Baga Gazaryn Chuluu | human | IEA, Xiongnu | 54 | 27.78 | 27.78 | 85.19 |
| Bayanbulag | human | Han | 14 | 92.86 | 92.86 | 100.00 |
| Bor Ovoo | human | LBA | 2 | 50.00 | 50.00 | 100.00 |
| Burgast | animal | LBA | 13 | 0.00 | 69.23 | 84.62 |
| Chandman Khar | human | Xiongnu, ME, unknown | 5 | 80.00 | 80.00 | 100.00 |
| Dunde Orontso cemetery | human | Xiongnu | 1 | 100.00 | 100.00 | 100.00 |
| Egiin Gol | plant | Xiongnu | 8 | 25.00 | 75.00 | 87.50 |
| Egiin Gol | human | LBA, Xiongnu | 21 | 57.14 | 61.90 | 66.67 |
| Emeelt Tolgoi | human | LBA | 4 | 25.00 | 50.00 | 75.00 |
| Gold Mod 2 | animal | LBA | 11 | 90.91 | 90.91 | 100.00 |
| Gold Mod 2 | human | LBA | 3 | 100.00 | 100.00 | 100.00 |
| Hets Mountain cave | animal | unknown | 2 | 0.00 | 0.00 | 0.00 |
| Hets Mountain cave | human | ME | 3 | 66.67 | 66.67 | 66.67 |
| Houvsgol | human | LBA | 5 | 0.00 | 0.00 | 100.00 |
| Khanuy valley | animal | LBA, Xiongnu | 38 | 78.95 | 78.95 | 100.00 |
| Khosvgol (A,B,C,D,F) | animal | LBA, unknown | 3 | 0.00 | 0.00 | 0.00 |
| Khosvgol (A,B,C,D,F) | human | LBA | 16 | 0.00 | 6.25 | 6.25 |
| Khushuut | human | LBA | 3 | 0.00 | 33.33 | 33.33 |
| Khyar Kharaach | human | LBA | 3 | 66.67 | 100.00 | 100.00 |
| Maikhan Tolgoi | animal | LBA | 2 | 0.00 | 0.00 | 0.00 |
| Maikhan Tolgoi | human | LBA | 9 | 44.44 | 44.44 | 44.44 |
| Orkhon valley (Moiltyn Am, Orkhon 1, 7) | animal | palaeo | 2 | 0.00 | 50.00 | 50.00 |
| Shine Jinst | human | unknown | 1 | 100.00 | 100.00 | 100.00 |
| Tevsh | human | LBA | 2 | 0.00 | 0.00 | 0.00 |
| Ulaan Boom | human | LBA | 3 | 33.33 | 66.67 | 66.67 |
| Zuun Bel | human | ME, unknown | 6 | 66.67 | 66.67 | 83.33 |
| **Modern samples** | | | | | | |
| Baga Gazaryn Chuluu | animal | modern | 5 | 0.00 | 0.00 | 100.00 |
| Bayan Unjuul | animal | modern | 1 | 0.00 | 0.00 | 0.00 |
| Burgast | animal | modern | 1 | 100.00 | 100.00 | 100.00 |
| Egiin Gol | plant | modern | 5 | 60.00 | 100.00 | 100.00 |
| Khanuy valley | animal | modern | 3 | 100.00 | 100.00 | 100.00 |
| Maikhan Tolgoi | plant | modern | 7 | 57.14 | 57.14 | 57.14 |
| Orkhon valley (Moiltyn Am, Orkhon 1, 7) | animal | modern | 1 | 100.00 | 100.00 | 100.00 |

*BA: Bronze Age, LBA: Late Bronze Age, EIA: Early Iron Age, ME: Mongol Empire.*

(Table 2). Only the animal sample collected from the Bayan Unjuul site falls outside the local (<50 km) $^{87}Sr/^{86}Sr$ range (Fig 6). Similar results are obtained with the EML isoscape (Fig S3 in S3 File.)

## Discussion

Baseline $^{87}Sr/^{86}Sr$ isoscapes are essential tools to study mobility and origin of past human populations. Here, we used RF and EML, two machine learning approaches aiming at producing a bioavailable $^{87}Sr/^{86}Sr$ isoscape locally-calibrated for Mongolia. Combining published bioavailable $^{87}Sr/^{86}Sr$ data [9,38] with plant data from 534 new sampling sites mostly from Arkhangai, in central Mongolia, we drastically improved the spatial prediction of the bioavailable $^{87}Sr/^{86}Sr$ compared to isoscapes that partially or completely encompassed Mongolia [9,38]. Both machine learning approaches led to similar isoscapes allowing us to explore the provenance of archaeological remains from multiple sites across Mongolia.

In Arkhangai, the $^{87}Sr/^{86}Sr$ measured in the plants closely aligned with the prediction of our isoscape, whereas the isoscape from Zazzo et al. [38], also generating using the RF approach, mainly predicts higher $^{87}Sr/^{86}Sr$ values at the sampling site than the plant values without correlating with the plant data (Fig 4). Zazzo et al. [38]'s isoscape was calibrated for the Altai region using samples from a very limited area. A comparison of both isoscapes (Fig S4 in S3 File) shows that $^{87}Sr/^{86}Sr$ predictions remain similar in the Altai region and in the northwest of Mongolia but decrease from 0.712–0.718 to 0.707–0.712 in central and eastern Mongolia. This decrease in predicted $^{87}Sr/^{86}Sr$ value comes with an overall improvement of the spatial error with errors below 0.001 SD in central Mongolia.

The isoscape reflects the complex geology of Mongolia. The ancient metamorphic and plutonic formation of the Sayan Mountains (Archean-Proterozoic), north of Mongolia, exhibits the highest $^{87}Sr/^{86}Sr$ values (> 0.714) and high spatial uncertainty. High $^{87}Sr/^{86}Sr$ values are generally associated with greater uncertainty, as plutonic formations such as granite tend to be more heterogeneous than sedimentary rocks, and the age estimates of older formations are typically less precise than those of younger ones [9,30]. The Mongol/Gobi Altai Mountain ranges and the Khangai Mountain ranges, formed by the collision of micro-continents and islandic arcs during the Paleozoic, presents intermediate $^{87}Sr/^{86}Sr$ values (0.711–0.714). Finally, lower and more precise $^{87}Sr/^{86}Sr$ values (< 0.711) are observed in younger sedimentary basins (Jurassic-Cretaceous) such as the Gobi Desert in southwest, or between the Altai and Khangai Mountain ranges. In the Arkhangai, the southwest-northeast gradient in $^{87}Sr/^{86}Sr$ illustrates the transition from the Khangai mountain to the carbonate-rich sedimentary steppes of the Bulgan and Töv aimags. Similar results are observed outside Mongolia with highly radiogenic regions in the Archean-Proterozoic Mountain ranges, associated with particularly high spatial uncertainty, in Russia and lower $^{87}Sr/^{86}Sr$ values in the sedimentary regions of China.

The RF model performed well, comparable to recently published isoscapes relying on the same method and a shared core bioavailable $^{87}Sr/^{86}Sr$ database [9,26,53]. Variables selected for the model aligned with previous work [9,26–29,53], though the importance-ranking mostly of low-ranked predictors varies due to the stochasticity of the RF approach [62]. Partial dependence plots further support this consistency, with notably the increase of the predicted $^{87}Sr/^{86}Sr$ values with the age and the predicted $^{87}Sr/^{86}Sr$ of the bedrock [9,26]. Local calibration was already effective with only 10% of the sampling sites (Fig 5). The addition of new samples led to a more gradual improvement of the model accuracy and precision up to 50% of sampling sites. The higher variability in RMSE at 90% was due to the fact that it was computed on a smaller dataset (10% of the sites). The sampling effort may have been more intensive than necessary given the 1 km resolution of the predictors used in the models. However, the Arkhangai plant dataset remains highly valuable for generating a high-resolution isoscape using kriging approach [22,23] to study mobility within Arkhangai.

The EML model performed similarly to the RF model but showed slightly lower agreement with plant $^{87}Sr/^{86}Sr$ values in Arkhangai. In a previous study, the EML algorithm implemented in the *landmap* package [63] improved isoscape precision compared to RF while keeping a similar distribution of the $^{87}Sr/^{86}Sr$ across the landscape [26]. Here, EML enhanced precision in radiogenic areas with high spatial uncertainty in the RF isoscape but mostly reduced elsewhere. EML appears to constrain extreme spatial error values [26], which benefits highly radiogenic and heterogenous regions such as old

cratonic formations [9,26], but may reduce precision in younger sedimentary formations with inherently low spatial uncertainty [9]. Despite broad similarities between the prediction of both models, $^{87}Sr/^{86}Sr$ values of both isoscapes differ in some regions, notably in old mountainous regions where EML predicted lower $^{87}Sr/^{86}Sr$ values. These differences between the RF and EML isoscapes occur mostly in areas of Mongolia that were not calibrated with local data and it is unclear which isoscape is more accurate. Additional sampling notably in areas of high uncertainty would help refine both isoscapes [9,26]. Until then, we recommend the RF isoscape for provenance and mobility studies within Mongolia due to its lower overall spatial uncertainty and better agreement with local plant values.

Published local animals and plant data, both modern and archaeological, are valuable for evaluating the accuracy of the isoscape in regions not included in our sampling campaigns. Among the 11 sites for which local animal and plant $^{87}Sr/^{86}Sr$ values were available, 8 sites showed values consistent with the prediction of the isoscape. At Maikhan Tolgoi [36], the plant data aligned with the isoscape prediction while animal data were slightly above but within the spatial error. Animal samples from Baga Gazaryn Chuluu [41], despite an average $^{87}Sr/^{86}Sr$ consistent with the local baseline, showed two distinct $^{87}Sr/^{86}Sr$ signatures that fall respectively slightly above and slightly below the $^{87}Sr/^{86}Sr$ range at 10 and 20 km. High values for argali (*Ovis Ammon*), foraging at high elevation, likely reflected the granitic geology of the massif [41]. In contrast, lower values in domestic sheep, grazing in valleys and surrounding steppes, fell below the mountain range predictions [41]. The isoscape, limited by the resolution and accuracy of the geological variables, possibly failed to capture fine scale geological variation in this area such as the low $^{87}Sr/^{86}Sr$ of the sedimentary valley and the high $^{87}Sr/^{86}Sr$ of the granitic ridges, and provide an averaged value. Finally, the 3 sites that did not align with the isoscape, Khösvgöl, in the North, Bayan Unjuul, south of Ulaanbaatar, and Hets Mountain cave, South-East close to the broader with China, presented local values from rodent data [37,77,78] well below the isoscape prediction either using RF or EML. However, local baseline values for these sites rely on very few rodent samples and might not reflect the full variation of the neighboring bioavailable $^{87}Sr/^{86}Sr$. On the other hand, the RF and EML were not trained on data from these regions and additional local modern samples would help to refine the isoscape prediction.

The comparison between the predicted bioavailable $^{87}Sr/^{86}Sr$ and human, animal and plant archaeological data from multiple sites across Mongolia allowed to explore the mobility of past human groups at different periods of Mongolia history. The averaged $^{87}Sr/^{86}Sr$ values estimated at each archaeological site suggest a predominantly local origin (within 50 km) for most sites (21 out of 25), regardless of the time period. These findings are consistent with the general conclusion of most of the studies comparing human remains to local $^{87}Sr/^{86}Sr$ baselines (Table S3). Looking more closely, more than half of the individuals were identified as local for 16 of the 20 sites with human samples, and at 5 of the 9 sites with animal remains. Only 9 sites yielded human material that was likely entirely local. For example, at the Kanuy valley site, despite an overall local signature of their samples, Makarewicz et al. [45] identified 3 out of 19 non-local horses. At Egiin Gol, Macichek et al. [40] considered that 20 out of 21 of their human individuals were local but without relying on a real $^{87}Sr/^{86}Sr$ baseline. We identified their outlier (sample: 0.7128, $^{87}Sr/^{86}Sr$ local range: 0.7082–0.7101) but also 6 individuals that fell shortly below, although within the spatial error. In the Altai mountains, Zazzo et al. [38] relied on a similar isoscape to explore local mobility of Late Bronze Age horses around the Burgast archaeological site, using Bayesian spatial assignment [25]. They identified one non-local individual, presenting $^{87}Sr/^{86}Sr$ values in its early life (0.706–0.707) below Altai $^{87}Sr/^{86}Sr$ range (0.710–0.719), but also below the range of their isoscape, only calibrated with Altai data. Based on our isoscape this horse could be from the actual Arkhangai or Bulgan aimags, implying an origin more than 500 km away from Burgast. Among the sites for which our results did not align with the original study, our isoscape suggested a local origin at Bayanbulag, in southern Mongolia, in contrast with Cui et al. [78] who inferred a non-local origin for Han dynasty human remains. However, it is worth pointing that the authors relied on isotope values located 200 km away in the Hets Mountain Cave. Further discrepancies between our results and those of previous studies were observed at Khösvgöl [37], Bayan Unjuul [40] and Hets Mountain cave [78]. In these cases, non-local origin for these sites was supported by local $^{87}Sr/^{86}Sr$ values from local animal samples (rodents), suggesting a problem of accuracy for the isoscape in these regions. Our

results suggest that most of the buried humans and animals exhibit a local origin. Their movements throughout life appear to have been largely confined to a limited area, typically within a few tens of kilometers. While some individuals may have traveled longer distances, such movements were likely not sustained and/or frequent enough to be significantly recorded in their tissues. These findings align with current knowledge of human and livestock mobility, particularly in central Mongolia, where nomadic routes tend to follow geographically restricted patterns [81]. It is therefore plausible that the mobility patterns observed in the archaeological record reflect those of present-day mobile pastoralist communities. However, the current dataset does not adequately capture long-term temporal variability or the full geographic and cultural diversity of the Mongolian territory. Under these circumstances, proposing broad-scale models of mobility would be premature. Only micro-regional studies—focused on individual sites or closely clustered site groups—can, at this stage, support robust interpretations. This is exemplified by cases where long-distance mobility or non-local origins have been detected, such as at Burgast [38] or in the Khanuy Valley [45]. These examples suggest that it may be possible to identify specific individuals more inclined to undertake extended movements. It is worth pointing that we adopted a coarse approach by comparing archaeological data pooled at the site level with the isoscape prediction. More refined methods, such as Bayesian spatial assignment, would allow to investigate individual origin of each sample with greater precision [11,25,38].

## Conclusion & perspective

In this paper we present the first bioavailable $^{87}Sr/^{86}Sr$ isoscape calibrated locally for Mongolia. As a simple application, we used this isoscape to assess broadly the local status of archaeological remains from different sites, identifying a local origin of the samples for most of the sites in agreement with previous studies. Prediction maps, along with their spatial uncertainty, enables probabilistic assignment analyses [25]. When combined with high-resolution $^{87}Sr/^{86}Sr$ sampling using Laser Ablation Multi-Collector Inductively Coupled Plasma Mass Spectrometry [14,38], it provides a powerful tool for investigating the origin and mobility of individuals [14,38]. Moreover, Mongolia presenting a North-South gradient in $\delta^{18}O$ [82], assignments could be refined using a multi-isotope approach incorporating $\delta^{18}O$ with $^{87}Sr/^{86}Sr$ to help discriminating between several likely regions of origin [11]. This, in turn, would greatly improve our understanding of past mobility in Mongolia and would offer new insights into how mobility shaped Mongolian society through time. The bioavailable $^{87}Sr/^{86}Sr$ isoscape of Mongolia has also potential applications beyond the archaeological field. Since $^{87}Sr/^{86}Sr$ reflects early life movements of animals, this method has been used in ecology to track the migratory patterns of birds and mammals [6,83,84]. In Mongolia, it could help monitor the mobility of migratory or nomadic species, such as the Mongolian gazelle (*Procapra gutturosa*) and the critically endangered saiga antelope (*Saiga tatarica*). Furthermore, $^{87}Sr/^{86}Sr$ has been applied in food authentication and traceability, ensuring the provenance of agricultural products [85,86]. This isoscape could serve as a valuable tool for authenticating the origin of high-end Mongolian products, such as cashmere wool. By establishing a robust baseline for provenance studies in Mongolia, this isoscape also represents a significant contribution to $^{87}Sr/^{86}Sr$ mapping in Asia, a region far less documented than Europe and North America [9]. Future studies focusing on under-sampled areas will be essential to refining this isoscape at both local and broader scales, ultimately enhancing our understanding of Asia's isotopic landscape.

## Supporting information

**S1 Table. Supplementary material SM1_Table S1: Bioavailable $^{87}Sr/^{86}Sr$ database.** Includes soil, plant and animal samples used for the random forest and the ensemble machine learning analyses.
(XLSX)

**S2 Table. Supplementary material SM2_Table S2: Archaeological sites from Mongolia with published archaeological and modern $^{87}Sr/^{86}Sr$ data.** Coordinates in italic were approximated from the maps of the source studies when longitude and latitude information were missing or were averaged when multiple sites were excavated within the study area.

BA: Bronze Age, LBA: Bronze Age, EIA: Early Iron Age, ME: Mongol Empire. Unknown period corresponds to archaeological samples with unspecified age.
(XLSX)

**S3 File. Supplementary material SM3: Fig S1 to S4.**
(PDF)

**S4 File. Supplementary material SM4: R script to generate the bioavailable $^{87}$Sr/$^{86}$Sr isoscape of Mongolia.**
(TXT)

**S5 File. Supplementary material SM5: RF isoscape with its associated spatial uncertainty (standard deviation) map.**
(ZIP)

**S6 File. Supplementary material SM6: RF isoscape with its associated spatial uncertainty (standard deviation) map.**
(ZIP)

**S7 File. Inclusivity in global research.**
(DOCX)

## Acknowledgments

We thank Lisa Garbé for the contribution in the preparation at the MNHN of the modern plants collected between 2019 and 2021. We thank Amélie Ogier for her contribution in the preparation at the LSCE of the last set of modern plants collected outside Arkhangai in 2023–2024. Measurement of the $^{87}$Sr/$^{86}$Sr ratios used for the isoscape was carried out at LSCE using PANOPLY's MC-ICPMS facilities.

## Author contributions

**Conceptualization:** Mael Le Corre, Sébastien Lepetz, Antoine Zazzo.

**Data curation:** Mael Le Corre.

**Formal analysis:** Mael Le Corre.

**Investigation:** Eric Douville, Arnaud Dapoigny, Khanh-Vi Tran, Tsagaan Turbat, Ganbold Enkhbayar, Sébastien Lepetz, Antoine Zazzo.

**Methodology:** Mael Le Corre, Eric Douville, Antoine Zazzo.

**Project administration:** Eric Douville, Antoine Zazzo.

**Resources:** Eric Douville, Tsagaan Turbat, Antoine Zazzo.

**Software:** Mael Le Corre.

**Supervision:** Eric Douville, Antoine Zazzo.

**Validation:** Mael Le Corre, Eric Douville, Arnaud Dapoigny, Khanh-Vi Tran.

**Visualization:** Mael Le Corre.

**Writing – original draft:** Mael Le Corre, Sébastien Lepetz, Antoine Zazzo.

**Writing – review & editing:** Eric Douville, Arnaud Dapoigny, Khanh-Vi Tran, Tsagaan Turbat, Ganbold Enkhbayar, Sébastien Lepetz, Antoine Zazzo.

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
