## [Decision Letter · Decision Letter 0]

20 Aug 2025

PONE-D-25-32694A bioavailable 87Sr/86Sr isoscape of Mongolia: implications for the reconstruction of past human and animal mobilityPLOS ONE

Dear Dr. Le Corre,

Thank you for submitting your manuscript to PLOS ONE. After careful consideration, we feel that it has merit but does not fully meet PLOS ONE’s publication criteria as it currently stands. Therefore, we invite you to submit a revised version of the manuscript that addresses the points raised during the review process. 

Both reviewers recommend minor revisions to your manuscript. I believe the paper is a solid and well-executed piece of scientific work, and I likewise recommend only minor revisions.

My only comment to the authors is to carefully review the recent literature on Sr isoscapes, as other studies have applied machine learning models in this context and may warrant acknowledgment.

We look forward to receiving your revised manuscript.

Kind regards,

Federico Lugli, Ph.D.

Academic Editor

PLOS ONE

Journal Requirements:

3. In your manuscript, please provide additional information regarding the specimens used in your study. Ensure that you have reported human remain specimen numbers and complete repository information, including museum name and geographic location.

For more information on PLOS One's requirements for paleontology and archeology research, see https://journals.plos.org/plosone/s/submission-guidelines#loc-paleontology-and-archaeology-research.

“This research, including salary support for MLC and KVT, sample collection, and isotope analysis (LSCE), was funded by the French National Research Agency under grant number ANR-20-CE27-0018-02.”

5. Thank you for stating in your Funding Statement:

“This research, including salary support for MLC and KVT, sample collection, and isotope analysis (LSCE), was funded by the French National Research Agency under grant number ANR-20-CE27-0018-02.”

6. We noted in your submission details that a portion of your manuscript may have been presented or published elsewhere. [This manuscript includes previously published ⁸⁷Sr/⁸⁶Sr data from archaeological sites (Table 2, Table S2, Figure 6), which are compared to the isoscape predictions as an example of application. While several of these studies also aimed to assess local vs. non-local origins of individuals, they relied on point-based local baselines (e.g., fauna, plants) specific to each site. In contrast, our study uses these same ⁸⁷Sr/⁸⁶Sr values to evaluate a novel, spatially continuous isoscape model. The aim is not to reinterpret the archaeological data per se, but to demonstrate the potential of the isoscape for assessing the origin of archaeological material and to evaluate the isoscape’s predictive accuracy. As such, this does not constitute dual publication.] Please clarify whether this [conference proceeding or publication] was peer-reviewed and formally published. If this work was previously peer-reviewed and published, in the cover letter please provide the reason that this work does not constitute dual publication and should be included in the current manuscript.

7. We note that Figures 1 & 2,  S1 & S2 in your submission contain [map/satellite] images which may be copyrighted. All PLOS content is published under the Creative Commons Attribution License (CC BY 4.0), which means that the manuscript, images, and Supporting Information files will be freely available online, and any third party is permitted to access, download, copy, distribute, and use these materials in any way, even commercially, with proper attribution. For these reasons, we cannot publish previously copyrighted maps or satellite images created using proprietary data, such as Google software (Google Maps, Street View, and Earth). For more information, see our copyright guidelines: http://journals.plos.org/plosone/s/licenses-and-copyright.

a. You may seek permission from the original copyright holder of Figures 1 & 2 ,S1 & S2 to publish the content specifically under the CC BY 4.0 license. 

8. We are unable to open your Supporting Information file [SM4_R_script_isoscape.zip]. Please kindly revise as necessary and re-upload.

Reviewers' comments:

Reviewer's Responses to Questions

**Comments to the Author**

1. Is the manuscript technically sound, and do the data support the conclusions?

Reviewer #1: Yes

Reviewer #2: Yes

2. Has the statistical analysis been performed appropriately and rigorously? 

Reviewer #1: Yes

Reviewer #2: Yes

3. Have the authors made all data underlying the findings in their manuscript fully available?

Reviewer #1: Yes

Reviewer #2: Yes

4. Is the manuscript presented in an intelligible fashion and written in standard English?

Reviewer #1: Yes

Reviewer #2: Yes

5. Review Comments to the Author

Reviewer #1: The authors present a wonderful study: 1) the goals are well-defined; 2) the methodological approach is robust, formally exploring alternative machine learning algorithms (RF and EML) to build strontium isoscapes; 3) the results are extensive and clearly described and analyzed; and 4) the discussion is warranted by the baseline presented in this paper and the inferences are both conservative and interesting. An additional merit of the paper is the public availability of all the data, analytical procedures and codes utilized to develop the paper. Hence, this research is fully and easily replicable.

I don't have signficant suggestions and strongly recommend its publication in PLos One.

Reviewer #2: General comments

This paper presents an impressive number of 87Sr/86Sr samples for an understudied region of the world. This contributes greatly to the ongoing efforts to create bioavailable strontium isoscapes around the world, thank you! Thank you also for providing the variables in a downloadable format as well as all the code used in this study.

The manuscript itself was easy to read and has a nice clear structure. The thorough explanation of the modelling methods employed is appreciated. I have only minor comments (specific ones below), and only have a few slightly larger (but optional suggestions) to the authors. These larger suggestions are more interest on my account that requirements for the paper!

In the Introduction (lines 132 – 152) as you can’t see the dates of references it is a little confusing the order of the environmental data collection, perhaps add the order of these studies into the text.

In Methods (line 228), you mention that notably the predicted bedrock 87Sr/86Sr is included. Why is that notable and what difference can this potentially make to the model output?

In Methods (line 279), you state that the EML model accounts for spatial auto-correlation between samples. My questions for this statements were does spatial autocorrelation hold for Sr given distinct boundaries between geological units? Does the appearance (or not) of spatial autocorrelation influence the results of EML model? RF does not take into account the closeness of sampling sites, but does EML? Consider adding a sentence or two addressing this, if it makes sense!

In Results, (lines 307 onwards), it isn’t very clear which data is being used for each of the model iterations, consider making that clearer.

Specific comments

Line 93, line 136, line 143, line 233 – Capitalise the Sr on 86Sr

Line 110 – perhaps define where is High and Central Asia

Line 145 – capitalise B on Bronze Age

Line 156 – “exceeding locally 4000m” is odd phrasing, perhaps “Mongolia has an average elevation of 1500 m, with some regions exceeding 4000 m.”

Figure 1 – a lot of specific regions/mountain ranges etc are mentioned in the preceding and later paragraphs, perhaps add these to Fig 1 to make it clearer for the reader to follow. The blue dots are also a little difficult to see, perhaps a slightly darker blue would be easier?

Line 253 – should it be tree not three?

Line 487 – capitalise the Sr on 87Sr

Line 318 - Perhaps “but reaches RMSE = “ instead of “but allows to reach RMSE”

Line 319 – Perhaps reword to “The EML has similar performance compared to the RF model”

Line 377 – over is not needed in “spanning over less”

Line 504 – “without relying on a real” instead of relying to

6. PLOS authors have the option to publish the peer review history of their article (what does this mean? ). If published, this will include your full peer review and any attached files.

**Do you want your identity to be public for this peer review?** For information about this choice, including consent withdrawal, please see our Privacy Policy .

Reviewer #1: **Yes: ** Ramiro Barberena

Reviewer #2: No

---

## [Author Response · Author response to Decision Letter 1]

3 Oct 2025

Responses to comments:

We are very grateful to the reviewers for their thoughtful and positive evaluations of our work. Their comments and suggestions have been extremely valuable in revising the manuscript, and we have addressed each point in detail below. To facilitate your evaluation of our revision, we have numbered and copied below the comments of each reviewer and have inserted our answers (preceded by stars) within the reviewers’ text. Line numbers refer to the revised version of the manuscript without track changes. Our responses to reviewer’s comments are followed by the journal requirements list.

Editor:

1) My only comment to the authors is to carefully review the recent literature on Sr isoscapes, as other studies have applied machine learning models in this context and may warrant acknowledgment.

*** We updated our reference list with several recent Sr isoscape studies using random forest analyses:

Armaroli E et al. 2024. Spatial ecology of moose in Sweden: Combined Sr-O-C isotope analyses of bone and antler. PLoS One.

Wang X, et al. 2024. Strontium isoscape of sub-Saharan Africa allows tracing origins of victims of the transatlantic slave trade. Nat Commun.

Scaggion C, et al. 2025. Random forest-based bioavailable strontium isoscape for environmental and archaeological applications in central eastern Argentina and western Uruguay. PLoS One.

Reviewer #1:

2) The authors present a wonderful study: 1) the goals are well-defined; 2) the methodological approach is robust, formally exploring alternative machine learning algorithms (RF and EML) to build strontium isoscapes; 3) the results are extensive and clearly described and analyzed; and 4) the discussion is warranted by the baseline presented in this paper and the inferences are both conservative and interesting. An additional merit of the paper is the public availability of all the data, analytical procedures and codes utilized to develop the paper. Hence, this research is fully and easily replicable.

I don’t have significant suggestions and strongly recommend its publication in Plos One.

*** We warmly thank the reviewer for their very positive and encouraging comments on our work. We are pleased that our manuscript was found to be clear and well-grounded, and that the open availability of data and code was appreciated. We are grateful for the reviewer’s strong recommendation for publication.

Reviewer #2:

General comments

3) This paper presents an impressive number of 87Sr/86Sr samples for an understudied region of the world. This contributes greatly to the ongoing efforts to create bioavailable strontium isoscapes around the world, thank you! Thank you also for providing the variables in a downloadable format as well as all the code used in this study. The manuscript itself was easy to read and has a nice clear structure. The thorough explanation of the modelling methods employed is appreciated. I have only minor comments (specific ones below), and only have a few slightly larger (but optional suggestions) to the authors. These larger suggestions are more interest on my account that requirements for the paper!

*** We sincerely thank the reviewer for their very positive evaluation of our work. We also thank the reviewer for the constructive comments and suggestions, which we have carefully considered and addressed below

3.1) In the Introduction (lines 132 – 152) as you cant see the dates of references it is a little confusing the order of the environmental data collection, perhaps add the order of these studies into the text.

*** Studies are provided in their order of publication. We have revised the text to include explicit temporal markers in order to highlight the sequence of contributions.

3.2) In Methods (line 228), you mention that notably the predicted bedrock 87Sr/86Sr is included. Why is that notable and what difference can this potentially make to the model output?

*** The underlying lithology is the main driver of the 87Sr/86Sr values in the environment as Sr from bedrock is expected to be the main contributor to the bioavailable 87Sr/86Sr pool. The Sr bedrock model from Bataille et al. 2014 integrates the age and the nature of the bedrock to predict its 87Sr/86Sr value. We added these sentences lines 232-234 :

“Bedrock age and nature are the main driver of the 87Sr/86Sr variations in the landscape [8,9]. This predictive 87Sr/86Sr bedrock model integrates the main sources of geochemical variation that is expected to propagate into soils and consequently into the bioavailable 87Sr/86Sr pool [9,31]. “

3.3) In Methods (line 279), you state that the EML model accounts for spatial auto-correlation between samples. My questions for this statements were does spatial autocorrelation hold for Sr given distinct boundaries between geological units? Does the appearance (or not) of spatial autocorrelation influence the results of EML model? RF does not take into account the closeness of sampling sites, but does EML? Consider adding a sentence or two addressing this, if it makes sense!

*** We fully agree that for strontium, sharp boundaries between geological units can reduce spatial autocorrelation at local scales. In such contexts, the EML's predictive power derives more directly from environmental predictors (e.g., geological type, soil characteristics). Nonetheless, at broader spatial scales (regional to continental), geological units often span substantial areas, so spatial dependency remains informative and effectively captured by the model as the model is train on a global, worldwide dataset. EML does not rely on the distance between nearby sites; instead, it incorporates oblique geographic coordinates as covariates to capture broad-scale spatial trends, and applies spatial cross-validation to mitigate the effect of clustered samples. We added the following sentences to clarify this point lines 282-286 :

“Local auto-correlation may be weak due to sharp boundaries between geological units, with predictions relying mainly on environmental predictors. At broader spatial scales (regional or continental), however, geological units extend over large areas, and spatial dependency remains informative for the model [27]. Using oblique coordinates as covariate help capture these potential broad-scale spatial trends.”

3.4) In Results, (lines 307 onwards), it isnt very clear which data is being used for each of the model iterations, consider making that clearer.

*** The RF and EML isoscapes were generated using both the published global 87Sr/86Sr dataset and the new samples collected for this study. To assess the contribution of our sampling effort, we ran RF models that incorporated different amounts of the newly collected plant samples and evaluated how model performance and predictions improved. Accordingly, we reorganized Section “3.1 Model performance” to first present the RF and EML results based on the complete dataset (published data + our samples), followed by the assessment of model improvements obtained by progressively adding our new samples to the database.

Specific comments:

3.5) Line 93, line 136, line 143, line 233 – Capitalise the Sr on 86Sr

*** Corrections done.

3.6) Line 110 – perhaps define where is High and Central Asia

*** We added some precisions:

“The spatial distribution of bioavailable 87Sr/86Sr in Asia, notably in High Asia (Tibetan Plateau and surrounding mountain ranges) and Central Asia (Kazakhstan and neighboring steppe countries, including Mongolia), is poorly documented”.

3.7) Line 145 – capitalise B on Bronze Age

*** Correction done.

3.8) Line 156 – “exceeding locally 4000m” is odd phrasing, perhaps “Mongolia has an average elevation of 1500 m, with some regions exceeding 4000 m.”

*** We modified the sentence according to the reviewer ‘s suggestion.

3.9) Figure 1 – a lot of specific regions/mountain ranges etc are mentioned in the preceding and later paragraphs, perhaps add these to Fig 1 to make it clearer for the reader to follow. The blue dots are also a little difficult to see, perhaps a slightly darker blue would be easier?

*** Main mountain ranges of Mongolia were added to the map, blue dots were changed to purple dots.

3.10) Line 253 – should it be tree not three?

*** Correction done.

3.11) Line 487 – capitalise the Sr on 87Sr

*** Correction done.

3.12) Line 318 – Perhaps “but reaches RMSE = “ instead of “but allows to reach RMSE”

*** We modified the sentence according to the reviewer ‘s suggestion.

3.13) Line 319 – Perhaps reword to “The EML has similar performance compared to the RF model”

*** We modified the sentence according to the reviewer ‘s suggestion.

3.14) Line 377 – over is not needed in “spanning over less”

*** Correction done.

3.15) Line 504 – “without relying on a real” instead of relying to

*** Correction done.

---

## [Editor Report · Decision Letter 1]

23 Oct 2025

A bioavailable 87Sr/86Sr isoscape of Mongolia: implications for the reconstruction of past human and animal mobility

PONE-D-25-32694R1

Dear Dr. Le Corre,

We’re pleased to inform you that your manuscript has been judged scientifically suitable for publication and will be formally accepted for publication once it meets all outstanding technical requirements.

Kind regards,

Federico Lugli, Ph.D.

Academic Editor

PLOS ONE
---

## [Editor Report · Acceptance letter]

PONE-D-25-32694R1

PLOS ONE

Dear Dr. Le Corre,

I'm pleased to inform you that your manuscript has been deemed suitable for publication in PLOS ONE. Congratulations! Your manuscript is now being handed over to our production team.

Kind regards,

on behalf of

Dr. Federico Lugli

Academic Editor

PLOS ONE